# Sex Differences between Medical Students in the Assessment of the Fear of COVID-19

**DOI:** 10.3390/ijerph19063372

**Published:** 2022-03-13

**Authors:** Maria del Carmen Trapp, Brandt J. Wiskur, Joy H. Suh, Michael W. Brand, Katrin G. Kuhn, Julio Rojas

**Affiliations:** 1Department of Psychiatry and Behavioral Sciences, College of Medicine, University of Oklahoma Health Sciences Center, Oklahoma City, OK 73117, USA; maria-trapp@ouhsc.edu (M.d.C.T.); joy-suh@ouhsc.edu (J.H.S.); michael-brand@ouhsc.edu (M.W.B.); julio.rojas@potawatomi.org (J.R.); 2Academic Affairs and Faculty Development, University of Oklahoma Health Sciences Center, Oklahoma City, OK 73117, USA; 3Department of Biostatistics & Epidemiology, Hudson College of Public Health, University of Oklahoma Health Sciences Center, Oklahoma City, OK 73117, USA; katrin-kuhn@ouhsc.edu

**Keywords:** COVID-19, gender role, fear, anxiety, students

## Abstract

**Background:** Differing expressions of the fear of COVID-19 between men and women can potentially increase both immediate and long-term physical health risks. We predicted that women students would express greater fear of COVID-19. **Methods:** We used an Internet-delivered Fear of COVID-19 Scale (FCV-19S) to assess fear among men (*n* = 100) and women (*n* = 272) from a larger population of academic medical center members (*n* = 1761). Sex differences in emotional and physical symptoms were assessed as subcategories within fear scores. **Results:** Women reported greater fear of COVID-19 than men (*p* < 0.001). Women reported greater emotional fear (*p* < 0.001) on specific scale items (thinking of COVID-19, watching news stories about COVID-19, and losing sleep due to fear of contracting COVID-19). **Discussion/Conclusions:** These results provide a better understanding of how fear of COVID-19 can differ based on sex and how that fear may be expressed differently through emotional and physical symptoms. This information will inform academic health centers of COVID-19 prevention and management policies that may include a gender-specific focus.

## 1. Introduction

As of March 2022, the World Health Organization’s COVID-19 Dashboard has reported over 450 million cases and 6 million deaths globally attributable to COVID-19, with 954,913 deaths occurring in the United States [1]. Lockdowns, travel restrictions, and social distancing became essential to limiting the spread of COVID-19. However, these measures had severe economic consequences, such as the first U.S. recession since the Great Recession and sharply rising unemployment [2]. Financial insecurity and social isolation, alongside the stress of contracting COVID-19, exacerbated mental health complications worldwide. In Japan, monthly suicide rates increased 16% during the second wave of the pandemic from July to October 2020 [3]. A study in Hong Kong indicated that rates of depression and anxiety in the population are significantly higher than rates pre-pandemic [4]. 

The COVID-19 pandemic is negatively impacting psychological well-being. This negative impact on mental health is significant for the entire population but may have even worse consequences for populations with already poor mental health. Medical students are subjected to extreme stress conditions due to the significant challenges associated with their academic and clinical responsibilities. Medical students have higher rates of depression, anxiety, and psychological distress than the general population or age-matched peers [5]. COVID-19 adds stressors to these students. A survey of college students in the U.S. indicated that students have increased rates of depression and anxiety during COVID-19 due to stressors such as fear for the health of loved ones, decreased social interactions, and increased concern about academic performance [6]. Medical and other health professional students are likely susceptible to these same stressors, as their schools have transitioned to online learning and reduced in-person clinical opportunities during the pandemic [7]. Due to their already vulnerable mental health, it is important to quantify and study the impact of COVID-19 on the mental health of these students. As future health professionals, their mental health is essential for their education and future role as caretakers of the population.

COVID-19 has serious mental health implications for students [8]. Beyond the stress and anxiety resulting from lockdowns, many individuals fear simply contracting the virus [9]. This fear of COVID-19, as with other infectious diseases, creates psychosocial impacts such as stigmatization and discrimination [10]. High levels of fear may result in maladaptive responses and poorer mental health outcomes. To adequately assess the public’s fear of COVID-19, the Fear of COVID-19 Scale (FCV-19S) was developed and later verified in undergraduate, graduate, and medical student settings [11,12,13]. In this study, FCV-19S was used to measure the fear of COVID-19 within the students of an academic medical center. As women are playing an increasing role in the medical setting, our objective was to determine if one sex may have an increased need for mental health interventions. We, therefore, further analyzed differences in the fear of COVID-19 based on self-reported sex differences. The results from this study may identify specific subpopulations whose mental health needs should be addressed with specificity.

## 2. Methods

### 2.1. Sample and Data Collection Procedures

Sample and data collection were previously described [14]. Briefly, an Internet-delivered Qualtrics survey was used to collect responses from faculty, staff, and students at the University of Oklahoma Health Sciences Center between 21 May and 18 June 2020. The survey was distributed to approximately 3100 students, 1700 faculty, and 4300 staff members. The distribution timeframe coincided with the end of the spring semester and a telecommute requirement for nonessential university employees. Survey participant selection and potential sample size were limited to their current existence within the institutional email system and active enrollment in the spring semester for students; addresses were provided by the Office of Institutional Research. The University of Oklahoma Health Sciences Center Institutional Research Board approved the protocol (IRB #12034), and participants provided informed consent prior to beginning the survey. There was no penalty for withdrawing and no monetary compensation was associated with the study.

### 2.2. Survey

Participants responded to close-ended and multiple-choice questions, which included questions regarding demographic, employment, and predisposing health condition characteristics. The FCV-19S, a seven-item self-report questionnaire, was used to assess participants’ fear of COVID-19. Response values reported as strongly disagree (1 point), disagree (2 points), neither agree nor disagree (3 points), agree (4 points), and strongly agree (5 points). Questions responses are summed, and the overall fear assessment is ranked as low (<14), mild (>14 and <28), and high (>28). The itemized survey questions are “I am most afraid of COVID-19”, “It makes me uncomfortable to think about COVID-19”, “My hands become clammy when I think about COVID-19”, “I am afraid of losing my life because of COVID-19”, “When watching news and stories about COVID-19, I become nervous and anxious”, “I cannot sleep because I’m worried about getting COVID-19”, and “My heart races or palpitates when I think about getting COVID-19”. The Total Fear Score was further subcategorized into physical and emotional symptoms. Physical symptom questions include “My hands become clammy when I think about COVID-19”, “I cannot sleep because I’m worried about getting COVID-19”, and “My heart races or palpitates when I think about getting COVID-19”. Emotional questions include “I am most afraid of COVID-19”, “It makes me uncomfortable to think about COVID-19”, “I am afraid of losing my life because of COVID-19”, and “When watching news and stories about COVID-19, I become nervous and anxious”.

### 2.3. Data Analysis

The FCV-19S was originally developed by Ahorsu and has since been validated in multiple countries to include the United States [11,13,15]. SAS Enterprise Guide^©^ (Cary, NC, USA) was used for survey result analysis. The study sample was characterized by descriptive statistics. Internal consistency was verified by Cronbach’s alpha. Mean scores, standard deviations (SD), median, and inner-quartile values (IQR) of the survey questions were analyzed. Population groups were compared by non-parametric 1-way analysis of variance (ANOVA). The Wilcoxon signed-rank test was used to compare fear ratings based on participants’ gender. Kruskal–Wallis with Dwass, Steel, and Critchlow-Fligner multiple comparison (post hoc) tests was also performed.

## 3. Results

In Table 1, we report that 375 students responded to this study: 271 were women, 100 were men, and 4 preferred not to say. All student respondents agreed to the study and only those that reported as being a man or woman were included in the sex-comparison table. Students were subcategorized based on age, self-reported minority status, predisposing health condition (diabetes, chronic lung condition(s), immunodeficiency, severe obesity, chronic kidney disease, liver disease), smoking status, academic affiliation, occupational specialization, and a clinical or nonclinical professional health setting.

Correlation analysis by question is reported in Table 2. Cronbach’s alpha scores were greater than 0.80 for every question, with an overall score of 0.88. Sex-based group differences by item analysis included emotional and physical symptoms, and total scores. Significant differences are reported between men’s and women’s responses. Women were more fearful of COVID-19 than men in the total assessment of fear assessment score (*p* < 0.001). Women reported greater discomfort when thinking about COVID-19, greater anxiousness when watching news stories about COVID-19, and greater loss of sleep because they worry about contracting COVID-19.

Multivariate analysis showed greatest significance between sexes when women and men were subgrouped by care specialty (Allied Health and MD/OD/DDS/PA) and by clinical or nonclinical specialty (data not shown). Our results suggest that compared to men, women express greater overall fear of COVID-19. 

## 4. Discussion

Many studies have attempted to access the emotional impact of COVID-19 on U.S. healthcare workers, but COVID-19′s impact on medical and other health professional students has been less studied [13,16]. It is essential to study how COVID-19 impacts the mental health of these students, as they will soon become frontline healthcare workers. Moreover, the demographics of U.S. healthcare workers are changing, and medical schools reflect this shift. Over the past 40 years, the proportion of women attending medical school in the U.S. has steadily risen [17]. There are now more women than men medical students. This demographic shift is global. In the U.K., women have outnumbered men in medical school by 3:2 since 2007 [18]. Dental schools have similar trends to medical schools, while physician assistant and nursing schools are both predominantly women [19,20,21]. As demonstrated by the demographics of healthcare students, most frontline healthcare workers are, or will soon be, women. It is important that policies pertaining to COVID-19 are adjusted to meet the needs of the changing demographic. 

In this study, women reported more fear of COVID-19 than men, reflected by their higher total and emotional fear scores. The findings are consistent with other studies’ conclusions that medical students are experiencing stress due to COVID-19 and women students are more likely to experience stress, anxiety and depression [22,23]. Specifically, women reported greater discomfort when thinking about COVID-19, greater anxiousness when watching news stories about COVID-19, and greater loss of sleep from worrying about contracting COVID-19. These issues may be defined as more emotional in nature and therefore, perhaps more difficult to address with universal interventions. Sex-specific health risks may increase fears of COVID-19, as women experience worse health outcomes for asthma, diabetes, and myocardial infarctions and report more mentally unhealthy days per year than men [24]. Previous research has also found that women healthcare professionals report more stress symptoms than men [25,26,27]. Policies that offer teleworking options when possible, strict social distancing guidelines for in-person interactions, access to vaccinations, and mask mandates may help mitigate COVID-19 fears as well as interventions that deal specifically with the emotional components of fear that have been highlighted. These may include group support sessions, focused emotions sessions, and enhanced coping strategies for emotions and fears that become overwhelming. 

Differences in pain anticipation and pain-related disorders are observed between the sexes [28,29,30]. In this study, women reported a greater emotional response to the fear of COVID-19, but results did not reach significance in the physical symptom responses. Addressing a limitation of this study, future research that includes expanded pain-sensitivity measurements and a larger population may help determine if emotional and physical symptom responses are maintained beyond the initial pandemic fear. A sustained increase in fear-related, physical symptom responses to COVID-19 may lead to more pain-related disorders, including headache, fibromyalgia, and irritable bowel syndrome, within medical professions with higher proportions of women.

An additional limitation of this study was the higher proportion of women respondents compared to men, resulting in unequal sample size. The unequal men-to-women ratio observed in this study is a common observation of online and traditional survey collection methods, particularly among educated women [31]. Additionally, university faculty members’ socio-demographic characteristics are not entirely representative of general population characteristics, limiting conclusions to academic health sciences centers. As the survey was not linked to institutional records, respondent characteristics to include academic performance and the demographic characteristics associated with their major could not be reported. Additional studies designed to address methodological limitations may enhance the specificity of potential policy recommendations.

Further research could attempt to assess if women experience more fear or are simply more willing to report fear than their men counterparts. As previously discussed, in this study and a previous validation study of students, the vast majority of the respondents were women [13]. The addition of a personality scale may help determine if the acquisition of the fear response associated with COVID-19 correlates with personality characteristic differences previously observed and reported between the sexes [28,32]. Men who are infected with SARS-CoV-2 are more likely to suffer from serious morbidity and mortality, yet men report lower COVID-19 fear scores [33]. We also analyzed data on self-reported sex as a male–female binary, but in larger populations, some individuals may self-report as gender nonconforming, nonbinary, or transgender (in the gender minority). Future research with a larger sample size could assess fears of COVID-19 in the gender minority community. These individuals may be more susceptible to fear as a result of greater mental health challenges, as research shows that gender minorities have significantly higher rates of depression and anxiety than cisgender communities [28].

## 5. Conclusions

In this study, the FCV-19S was used to compare men and women healthcare students at an academic medical center. Sex-specific differences were found, with women students reporting more fear and emotional features of fear being most salient and divergent. This data may be used to influence future policy changes in academic healthcare centers, as more sex-specific policy changes may help mitigate mental illness among students and employees. The data and implications may also serve to drive the type, style, and format of the actual interactions and programs with students to attenuate their fears from influencing or becoming long standing mental health issues. However, our study is limited by sample size and study design, such as the majority women respondents and limited pain-sensitivity measurements. Further studies should obtain a larger sample size, expand pain-sensitivity measures, and attempt to ascertain if men are simply less willing to report fear.

## Figures and Tables

**Table 1 ijerph-19-03372-t001:** Demographic data.

Variables		*n* = 375
Gender, % (*n*)	Female	72.3 (271)
	Male	26.7 (100)
	Prefer Not to Say	1.0 (4)
Age, % (*n*)	Less than 21	1.9 (7)
	21–25 years old	44.0 (165)
	25–34 years old	40.8 (153)
	35–44 years old	7.5 (28)
	45–54 years old	4.0 (15)
	55–64 years old	1.6 (6)
	65–74 years old	0.3 (1)
Minority, % (*n*)	Yes	18.13 (68)
	No	80.53 (302)
	Prefer Not to Say	1.33 (5)
Predisposing Health Condition, % (*n*)	Yes	13.1 (49)
	No	86.1 (323)
	Prefer not to say	0.8 (3)
Smokers, % (*n*)	Yes	8.0 (30)
	No	91.2 (342)
	Prefer not to say	0.8 (3)
Health Care Specialty, % (*n*)	Academic	0.3 (1)
	Administrative	1.1 (4)
	Allied Health	17.6 (66)
	Biomedical Science Research	8.5 (32)
	MD/OD/DDS/PA	42.1 (158)
	Nursing	17.9 (67)
	Pharmaceutical	5.6 (21)
	Psychological or Social Services	1.1 (4)
	Public Health	5.6 (21)
	Support Services	0.3 (1)
Clinical or Nonclinical, % (*n*)	Clinical	72.3 (271)
	Non-Clinical	27.7 (104)

**Table 2 ijerph-19-03372-t002:** Scores for assessment of fear of COVID-19.

	Overall		Men (100)			Women (271)		
Question	Correlation	α	Mean (SD)	Median (IQR)	Min, Max	Mean (SD)	Median (IQR)	Min, Max
I am most afraid of COVID-19	0.78	0.87	2.5 (1.2)	(1–3)	1, 5	2.7 (1.1)	(2–4)	1, 5
It makes me uncomfortable to think about COVID-19	0.80	0.87	2.3 (1.2) ***	(1–3)	1, 5	2.7 (1.3) ***	(2–4)	1, 5
My hands become clammy when I think of COVID-19	0.72	0.88	1.4 (0.7)	(1–2)	1, 4	1.6 (0.8)	(1–2)	1, 5
I am afraid of losing my life because of COVID-19	0.69	0.87	1.8 (1.1)	(1–2)	1, 5	2 (1.1)	(1–3)	1, 5
When watching news stories about COVID-19, I become nervous and anxious	0.77	0.87	2.4 (1.2) ***	(1–4)	1, 5	3 (1.4) ***	(2–4)	1, 5
I cannot sleep because I am worried about getting COVID-19	0.74	0.87	1.5 (0.7) ***	(1–2)	1, 4	1.6 (0.8) ***	(1–2)	1, 5
My heart races or palpitates when I think about getting COVID-19	0.75	0.87	1.4 (0.6)	(1–2)	1, 3	1.7 (1)	(1–2)	1, 5
Emotional subscale	0.93	0.84	9.1 (3.8) ***	(6–11.5)	4, 19	10.4 (4) ***	(7–13)	4, 20
Physical symptom subscale	0.81	0.85	4.3 (1.8)	(3–5.5)	3, 11	4.9 (2.4)	(3–6)	3, 15
Total assessment of fear score	1.00	0.88	13.4 (5.1) ***	(9–17)	7, 28	15.3 (5.9) ***	(11–19)	7, 34

“***” indicates a *p* value < 0.001.

## Data Availability

Aggregated data presented in this study are available on request from the corresponding author. The data are not publicly available in compliance with IRB study protocol.

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
