# Peer review of "Sex Differences between Medical Students in the Assessment of the Fear of COVID-19"

_ijerph, 2022, doi:10.3390/ijerph19063372_

Round 1

Reviewer 1 Report

Dear authors and editor,

The manuscript titled "Gender Differences in the Assessment of the Fear of COVID-19" This is a descriptive cross-sectional study  assessing fear of Covid-19 in a population of  health students. The authors assess gender differences.

There are many minor and major issues I'd like the authors resolve.

Abstract

1-Add the study design to the abstract. Also, the authors can choose to add the study design to the title of the manuscript.

2-Change the keywords. Delete the words "gender" and "; student " .  Not found in the MeSH (Medical Subject Headings). Change to "Gender Role" and "Students

3- The title does not refer to medical students. Very generic title that does not indicate the study population.

4-The abstract does not comply with the journal's guidelines.

Introduction

5-Adequate. It is recommended to state the objective more clearly.

Materials and Methods

6-Study size: Explain how the study size was arrived at.  

Results

7-It is understood that the differences may be due to differences in the sample between men (n=100) and women (n=271). It is understood that the sample of women is larger in health professions, however the comparison should be as equal as possible. Therefore, it is recommended to calculate a sample of both groups.

Discussion

 8-It is recommended to add a section on limitations.

Conclusion

9-It is recommended that a concluding section be added. Conclusions should be given with some caution because of the sample size and the design of the study.

Reference:

8-Adequate

Author Response

Dear authors and editor,

The manuscript titled "Gender Differences in the Assessment of the Fear of COVID-19" This is a descriptive cross-sectional study  assessing fear of Covid-19 in a population of  health students. The authors assess gender differences.

There are many minor and major issues I'd like the authors resolve.

Reviewer 1: Thank you for your clear suggestions and guidance to improve this manuscript. We have responded to every suggestion made as described below.

Abstract

1-Add the study design to the abstract. Also, the authors can choose to add the study design to the title of the manuscript.

The authors were required to reduce the overall length of the abstract, but we did add a statement of which better framed the study design: “This study used an internet-based delivered Fear of COVID-19 Scale (FCV-19S) to assess fear among men (n = 100) and women (n = 272) from a larger population of academic medical center members (n = 1761).” We hope that our abstract modifications meet your study design recommendation without reducing the perceived strength of the abstract (limited to 175 words).

2-Change the keywords. Delete the words "gender" and "; student " .  Not found in the MeSH (Medical Subject Headings). Change to "Gender Role" and "Students

Changed: “COVID-19; gender role; fear; anxiety; students

3- The title does not refer to medical students. Very generic title that does not indicate the study population.

Changed: “Gender Differences Between Medical Students in the Assessment of the Fear of COVID-19”

4-The abstract does not comply with the journal's guidelines.

We reduced the abstract to 175 words and included a better-framed study design in response to recommendation #1.

Introduction

5-Adequate. It is recommended to state the objective more clearly.

Attempted to state objective more clearly: “As women are playing an increasing role in the medical health care setting, our objective was to determine if one gender may have an increased need for mental health interventions. We, therefore, further analyzed differences in the fear of COVID-19 based on self-reported gender differences. Results from this study may identify specific sub-populations whose mental health needs should be addressed with specificity.”

Materials and Methods

6-Study size: Explain how the study size was arrived at.  Minor modification to the “Sample and data collection procedures.

The authors modified/added statement to address reviewer suggestion: “The survey was distributed to approximately 3,100 students, 1,700 faculty, and 4,300 staff members. The distribution timeframe coincided with the end of the spring semester and a telecommute requirement for non-essential University employees. Survey participant selection and potential sample size were limited to their current existence within the institutional email system and active enrollment in the spring semester for students; addresses provided by the Office of Institutional Research.”

Results

7-It is understood that the differences may be due to differences in the sample between men (n=100) and women (n=271). It is understood that the sample of women is larger in health professions, however the comparison should be as equal as possible. Therefore, it is recommended to calculate a sample of both groups.

The authors concur with the assessment and propose that this limitation is best addressed in the added “limitations” paragraph (below) of the discussion also recommended by Reviewer 1. Briefly, the sample size is limited to size of the student respondents with the response ratio generally representative of the enrollment counts for the Spring 2021 semester (specifically, 953 males and 2,116 females; ref https://admissions.ouhsc.edu/Portals/1047/assets/documents/Reports/21SP_ENR_DEG_GENDER.pdf) [m/f ratio .45 population and .37 study results]. Kruskal-Wallis with Dwass, Steel, and Critchlow-Fligner multiple comparison (post-hoc) tests post-hoc analysis confirmed the statistical significance reported. Additionally, women (particularly, educated women) are more likely to respond to online surveys (statement and reference added to the limitations paragraph) (https://www.researchgate.net/publication/234742407_Does_Gender_Influence_Online_Survey_Participation_A_Record-Linkage_Analysis_of_University_Faculty_Online_Survey_Response_Behavior) as well as other traditional survey requests (references within link above). Reductions of current sample size would reduce the power analysis and potential impact of the study.

Discussion

 8-It is recommended to add a section on limitations.

We addressed one limitation within paragraph: "Addressing a limitation of this study, future research that includes expanded pain sensitivity measurements and a larger population may help determine if emotional and physical symptom responses are maintained past the initial pandemic fear."

We further added a limitations paragraph: “An additional limitation of this study was the higher proportion of women respondents compared to men resulting in unequal sample size. The unequal male-to-female ratio observed in this study is a common observation of online and traditional survey collection methods, particularly among educated women [31]. Additionally, University faculty members socio-demographic characteristics are not entirely representative of general population characteristics, limiting conclusions to academic health sciences centers. As the survey was not linked to institutional records, respondent characteristics to include academic performance and the demographic characteristics associated with their major could not be reported. Additional studies designed to address methodological limitations may enhance the specificity of potential policy recommendations.”

Conclusion

9-It is recommended that a concluding section be added. Conclusions should be given with some caution because of the sample size and the design of the study.

A conclusion section was added to the paper.

“In this study, the FCV-19S was used to compare men and women healthcare students at an academic medical center. Gender-specific differences were found, with women students reporting more fear, with emotional features of fear being most salient and divergent. This data may be used to influence future policy changes in academic healthcare centers, as more gender-specific policy changes may help mitigate mental illness amongst students and employees. The data and implications may also serve to drive the type, style, and format of the actual interactions/programs with students to attenuate their fears from influencing or becoming long standing mental health issues. However, our study is limited by sample size and study design, such as having majority women respondents and limited pain sensitivity measurements. Further studies should obtain a larger sample size, expand pain sensitivity measures, and attempt to ascertain if men are simply less willing to report fear.”

Reference:

8-Adequate

Added one reference that helped address the “limitations” recommendation. The change resulted in a renumeration of references 31-33.

Reviewer 2 Report

This manuscript is on a timely and useful topic.  The Fear of COVID-19 scale has been well-examined in the past two years but continues to be relevant.  The issue of gender and fear of COVID in medical workers is useful.  While there is some initial research on gender and FCV-19S, I believe that this study could be a useful contribution to the literature on this scale.  The manuscript is generally well-written. 

There are some issues that undercut the study's significance, however.

First, word choices create confusion and vagueness.  For instance, the sexes are said to "process" the fear of COVID differently.  What does it mean to "process" fear?  The study does not shed light on any cognitive or emotional process.  A different word choice would remove this confusion.  In other places the term used is "express" emotion, which seems more accurate.

Similarly, the manuscript refers to "visceral" differences and calls a subset of the scale "visceral."  It is not clear whether the scale actually measures physical visceral processes; it measures the subjects' self-reported physical symptoms.   If this term has been used previously, it is defensible for use due to the need for consistency in the literature.  However, that is not made clear.  If not, a more accurate term should be considered. 

Second, psychometric data should be presented on the subscales being used here.  Have these subscales been previously validated?  If so, that should be reported in a bit more detail.  If not, more detail is warranted data analysis section of the survey.  It is reported that  a Cronbach's alpha was calculated, for instance, but what was its value?

Finally, the title of the manuscript refers to gender differences but the study actually focuses on healthcare students and should be presented with greater specificity in the title.

Author Response

This manuscript is on a timely and useful topic.  The Fear of COVID-19 scale has been well-examined in the past two years but continues to be relevant.  The issue of gender and fear of COVID in medical workers is useful.  While there is some initial research on gender and FCV-19S, I believe that this study could be a useful contribution to the literature on this scale.  The manuscript is generally well-written. 

There are some issues that undercut the study's significance, however.

First, word choices create confusion and vagueness.  For instance, the sexes are said to "process" the fear of COVID differently.  What does it mean to "process" fear?  The study does not shed light on any cognitive or emotional process.  A different word choice would remove this confusion.  In other places the term used is "express" emotion, which seems more accurate.

The authors concur with the reviewer’s identification of how “process” was vaguely used. All previous uses of “process” in regards to fear have been changed to “express.”

Similarly, the manuscript refers to "visceral" differences and calls a subset of the scale "visceral."  It is not clear whether the scale actually measures physical visceral processes; it measures the subjects' self-reported physical symptoms.   If this term has been used previously, it is defensible for use due to the need for consistency in the literature.  However, that is not made clear.  If not, a more accurate term should be considered. 

The authors concur with the reviewer’s observation that a physical metric was not used to evaluate “visceral” pain. In a previous study related to this study and referenced by this study, Kano et al. measured visceral symptoms using phasic esophageal stimulation. We used no such measure, and it is more accurate to report “physical symptoms.” The term visceral has been changed to physical symptoms.

Second, psychometric data should be presented on the subscales being used here.  Have these subscales been previously validated?  If so, that should be reported in a bit more detail.  If not, more detail is warranted data analysis section of the survey.  It is reported that  a Cronbach's alpha was calculated, for instance, but what was its value?

The authors concur that greater clarity of the internal consistency measure of this survey should be reported. We have added a statement to the Results section: “Cronbach’s alpha scores were greater than 0.80 in every question with an overall score of 0.88.”

Finally, the title of the manuscript refers to gender differences but the study actually focuses on healthcare students and should be presented with greater specificity in the title.

The authors appreciate this recommendation and have since changed the title to: “Gender Differences in the Assessment of the Fear of COVID-19”  

Round 2

Reviewer 1 Report

The authors have responded to the recommendations indicated. They have acknowledged the limitations of the manuscript and have improved or clarified important aspects.

Kind regards.